# Efficient Broadband Light-Trapping Structures on Thin-Film Silicon Fabricated by Laser, Chemical and Hybrid Chemical/Laser Treatments

**DOI:** 10.3390/ma16062350

**Published:** 2023-03-15

**Authors:** Michael Kovalev, Ivan Podlesnykh, Alena Nastulyavichus, Nikita Stsepuro, Irina Mushkarina, Pavel Platonov, Evgeniy Terukov, Sergey Abolmasov, Aleksandr Dunaev, Andrey Akhmatkhanov, Vladimir Shur, Sergey Kudryashov

**Affiliations:** 1Lebedev Physical Institute, 119991 Moscow, Russia; 2School of Natural Sciences and Mathematics, Ural Federal University, 620000 Ekaterinburg, Russia; 3Laser and Optoelectronic Systems Department, Bauman Moscow State Technical University, 2nd Baumanskaya St. 5/1, 105005 Moscow, Russia; 4Department of Electronics, St. Petersburg State Electrotechnical University, ul. Professora Popova 5, 197022 St. Petersburg, Russia; 5R&D Center of Thin Film Technologies in Energetics, 194064 St. Petersburg, Russia; 6All-Russian Research Institute for Optical and Physical Measurements, 119361 Moscow, Russia

**Keywords:** surface microstructures, light-trapping, reflection coefficient, laser texturing, chemical etching

## Abstract

Light-trapping structures formed on surfaces of various materials have attracted much attention in recent years due to their important role in many applications of science and technology. This article discusses various methods for manufacturing light-trapping “black” silicon, namely laser, chemical and hybrid chemical/laser ones. In addition to the widely explored laser texturing and chemical etching methods, we develop a hybrid chemical/laser texturing method, consisting in laser post-texturing of pyramidal structures obtained after chemical etching. After laser treatments the surface morphology was represented by a chaotic relief of microcones, while after chemical treatment it acquired a chaotic pyramidal relief. Moreover, laser texturing of preliminarily chemically microtextured silicon wafers is shown to take five-fold less time compared to bare flat silicon. In this case, the chemically/laser-treated samples exhibit average total reflectance in the spectral range of 250–1100 nm lower by 7–10% than after the purely chemical treatment.

## 1. Introduction

The fabrication and application of surface nano- and microstructures in modern technologies of photonics opens up new opportunities in such areas as information and communication technologies [1], solar photovoltaics [2], fabrication infrared (IR) silicon photoelements [3], molecular detection in chemistry and biomedicine [4]. At the same time, the field of micro- and nano-structuring of the surface of materials under the influence of nano-, pico-, and femto-second radiation with variable energy density and exposure due to the obvious advantages of laser pulses, namely, their short duration and high peak power, is being actively explored [5]. Therefore, in the last two decades, a certain interest has been attracted by experimental studies on the creation by such methods of not only hologram [6], diffraction [7], plasmonic [8], but also light-trapping structures [9,10], which make it possible to arbitrarily change the reflection coefficient of the metallic [11,12] and semiconductor [13,14,15,16,17] surfaces. In addition, such structures are capable of providing superhydrophobicity or superhydrophilicity [18], forming anticorrosion protection [19], etc. However, the use of such light-trapping textures, for example, in photovoltaics, as an alternative to antireflection coatings [20], rear emitter structures [21] and absorbing nanoparticles [22], makes it possible to increase the efficiency of solar cells (SCs) [2].

At present, the formation of a light-trapping microrelief on the surface of silicon SCs occurs chemically [23], due to the speed of processing, the low cost of reagents and the scalability of this technology to commercial production. However, the microstructure obtained on the surface of the material, fabricated chemically, as shown in [24], makes it possible to reduce the total reflection coefficient only to 12% in the silicon sensitivity wavelength range (250–1100 nm). In this case, with laser processing methods, the achieved reflection of the surface is a few percent [25,26]. In addition, most of the works describe the formation of structures by laser in the gas environment—sulfur hexafluoride [27,28,29], argon [30]—which greatly complicates the technology of material surface treatment. In this case, a decrease in the reflection coefficient of the silicon surface, as was shown in [31], made it possible to increase the photocurrent from the modified regions by 30%. Despite the improvement in the electrical characteristics of laser-structured SCs, the processing of wafers using pulsed radiation takes several hours, while chemical texturing takes several minutes.

In this article, we develop a general strategy for the hybrid chemical/laser fabrication of light-trapping structures on a silicon surface in air environment, which makes it possible to quickly and efficiently modify the surface of this material, and also compare this type of texturing with the common laser and chemical ones. We demonstrate that the designed structures provide highly efficient broadband (UV–near-IR, 250–1100 nm) antireflection of light, where the average total reflection coefficients of 4.4%, 15.0% and 7.9% were achieved on silicon surfaces processed by laser texturing, chemical etching and hybrid chemical/laser fabrication of structures, respectively. In addition, we point out that by laser refining the chemically textured surface of silicon wafers, better output characteristics could be achieved in photovoltaic applications.

## 2. Materials and Methods

As samples, monocrystalline n-type (phosphorus-doped) silicon wafers grown by the Czochralski method (CZ) were used: crystallographic orientation 〈100〉, resistivity 1–3 Ohm·cm; thickness 150 μm; size 157 × 157 mm^2^. Subsequently, the following variants of processing the obtained samples took place: laser microtexturing of the silicon wafer surface (sample #1); chemical etching of the silicon wafer surface in order to produce pyramidal structures (sample #2); hybrid chemical-laser processing consisting in laser post-texturing of pyramidal structures obtained after chemical etching (sample #3). For better understanding, the numbers of samples and corresponding processing methods are shown in a Table 1. Additionally a reference sample of silicon without any surface treatment (as cut wafer; untreated Si) participated in the studies on the characterization of the treated surface of materials.

A TETA femtosecond laser system (Avesta Project, Moscow, Russia) based on Yb^3+^ ions with a central wavelength of 1030 ± 3 nm and a maximum pulse repetition rate of 1 MHz acted as a radiation source. The beam quality parameter M^2^ was less than 1.25, the 1/e^2^-intensity laser beam diameter of the TEM_00_ mode is 5.0 ± 0.5 mm. The experiment used laser pulses with a duration of 250 fs and a maximum energy of 1 mJ. A ATEKO galvanoscanner (ATEKO, Moscow, Russia) equipped with a F-Theta lens, possessing a focal length of 100 mm, was used in a combination with a 3D motorized translation stage (Standa, Vilnius, Lithuania) to focus and scan the laser beam in three directions. The Rayleigh length of the transformed beam was 160 mm. Experiments on laser processing were carried out under laboratory conditions at normal incidence of the laser beam in an air environment. At the beginning of the study, small bands were textured in order to select the optimal processing parameters. During the experiments, the following processing parameters varied: the laser scanning speed varied from 0.125 mm/s to 300 mm/s, the average radiation power from 0.01 to 0.3 W, the pulse repetition rate from 1 to 300 kHz, and the diameter focusing of a Gaussian laser beam on 1/e^2^ of its maximum intensity from 20 to 140 μm. After selecting the optimal processing parameters, the microstructure was formed on a square area 25 × 25 mm^2^ in size. Scanning consisted in texturing lines of equal thickness along one direction, separated from each other by a certain distance—the scanning step. In this case, in the experiments the filling of the region was studied depending on the step, which varied in the range from 10 μm to 100 μm. After laser irradiation, all samples were cleaned in an ultrasonic bath with deionized water and dried with compressed air.

The chemical treatment of the samples was carried out by sequentially immersing silicon wafers in baths with various reagents. In addition, before and after each of the stages, the samples were cleaned in deionized water using ultrasonic baths, as in the case of laser texturing. First, the damaged layer was removed from the surface of the plates in a 15% KOH alkali (CAS No: 1310-58-3) solution heated to 80 °C. In this case, the etching depth of the plates was about 7 μm on each side. The fabrication of a pyramidal microstructure also took place in an alkaline solution of KOH at a temperature of 80 °C, but at a reduced concentration of KOH.

The textured surface morphology was analyzed using scanning electron microscopy (SEM; VEGA, TESCAN, Brno, Czech Republic). The SE detector was used as a receiver of secondary electrons. The accelerating voltage was 10 keV, the beam current was 100 pA, and the approximate beam spot was 30 nm. The atomic composition of the treated surface of the samples was determined by energy dispersive X-ray spectroscopy using an EDX detector mounted on SEM (Xplore EDX detector; Oxford Instruments, High Wycombe, UK). The spectral total reflectance in the wavelength range from 250 nm to 1100 nm was measured using a Perkin Elmer Lambda 900 spectrophotometer equipped with a PELA-100 integrating sphere at the spectral resolution of 5 nm [32]. The spectral transmittance of the samples was measured on a Perkin Elmer Lambda 950 spectrophotometer with a spectral resolution of 2 nm. All spectral parameters, such as reflectance and transmittance, were measured 5 times.

The crystalline state of each of the samples was analyzed using a 3D scanning laser confocal Raman microscope with a Confotec 350 spectrometer (SOL instruments, Minsk, Belarus) with a spectral resolution of 1.5 cm^–1^ and an excitation wavelength of 532 nm.

## 3. Results and Discussion

Figure 1 shows optical images of the surface of processed silicon samples at angles of −30°, 0° and 30°. In addition, Appendix A contain videos demonstrating the treated surface in the range of angles from −60° to 60° with step 10°. It is also worth noting the presence of white microdispersed particles (Figure 1a), which appeared as a result of cleaning sample #1. Such a defect can be caused by the wrong choice of the liquid in which the cleaning took place. During laser processing, the average power of the applied pulses and the focusing spot were 0.11 W, 70 µm (Figure 1a) and 0.024 W, 20 µm (Figure 1c) for samples #1 and #3, respectively; pulse repetition rate—1 kHz, 240 kHz. The scanning speed and step were 0.5 mm/s, 60 µm for sample #1 and 300 mm/s, 10 µm for sample #3. It should be noted that during laser texturing of sample #3 compared to sample #1, despite the lower applied radiation power and smaller focusing spot, the processing time was reduced by a factor of 5 (the processing time regarding the square area of 25 × 25 mm^2^ in size) due to the high scanning speed, which may indicate the possibility of improving the technology of hybrid chemical/laser fabrication of structures for ultrafast processing of the silicon surface using, for example, cylindrical lenses [33]. In addition, such a difference in processing time is associated with the presence of a previously chemically formed relief, on which laser processing is much faster. Optical images of the surfaces of samples #1 and #3 (Figure 1a,c), obtained using laser radiation, clearly show the coloring of the silicon surface in black. In this case, the surface of the chemically treated sample #2 (Figure 1b) has a characteristic blue tint, which may indicate a significantly greater reflection of the short-wavelength region of the visible spectrum than that of samples #1 and #3.

SEM images of the obtained structures are shown in Figure 2. The morphology of the laser-treated surface (samples #1, #3; Figure 2a,c) was a chaotic relief of microcones with various irregularities on their side faces. This type of relief is often called as self-organized structures, and the mechanism of its formation has been widely studied in many works, for example [34]. Based on the obtained images, it can be judged that there were no areas in which no structure was formed, which indicates good filling and the optimal choice of the scanning step during laser texturing. Moreover, sample #3 had a better textural homogeneity (i.e., the sizes of microcones remain constant) compared to sample #1. In both cases, the distance between the peaks of the structure was approximately 3–7 µm, their height varied from 8 to 10 µm. In contrast to the laser modification after chemical treatment (sample #2, Figure 2b), the resulting structure was a randomly arranged pyramid with a base width of 1 to 5 μm. The side faces did not have visible irregularities, and their orientation was parallel to the crystallographic planes 〈111〉 [35].

Table 2 presents the results of the atomic composition EDX analysis of the three samples. It should be noted that during the laser processing, the presence of oxygen is observed, which may indicate the formation of oxides during texturing. During chemical treatment, oxygen appears in the surface textures in a small amount. However, during the laser texturing in the air environment such high oxygen content is higher (Table 2). Its content can be significantly reduced by texturing in inert-gas environments.

The results of crystallographic analysis are shown in Figure 3. One graph shows the Raman spectra of all these three treated samples, as well as the untreated silicon. The four wafers under study were characterized by a sharp peak at 520 cm^−1^, which corresponds to the crystalline phase of silicon (c-Si). In addition, there was no peak at the Raman shift of 470 cm^−1^, which should correspond to amorphous silicon (α-Si). In this case, a slight smoothing of the left boundary of the crystalline silicon peak was observed, which may indicate an insignificant presence of an amorphous phase in the samples. On the basis of the obtained data on the samples, it can be concluded that, after treatment, silicon appears in the form of polycrystals (poly-Si) [36].

Figure 4 shows the spectra of the total reflectance of the silicon surface in the wavelength range from 250 nm to 1100 nm. According to the results of measuring the transmittance of silicon plates, all three modified samples turned out to be opaque in the same spectral range, and only for untreated silicon transmittance increases from 0% to 4% in the wavelength range from 900 nm to 1100 nm. Subsequently, the absorptance was calculated (Figure 5) using the formula:A = 1 − R − T,(1)
where A—absorptance, R—total reflectance, T—transmittance. In Figure 4 and Figure 5 the derived R and A values were presented as a percentage.

As a result of modification by various methods, micro- and nanostructures were obtained, which significantly reduced the reflection coefficient of the silicon surface. Laser treatment (sample #1) provided an average total reflection coefficient in the studied spectral range of 4.4%, chemical treatment (sample #2)—15.0%, chemical/laser treatment (sample #3)—7.9%. The average reflection of untreated silicon was about 39.4%. In this case, the minimum reflection coefficients for sample #1 were obtained—3.5% at a wavelength of 680 nm, for sample #2—8.9% at 960 nm, and for sample #3—6.9% at 800 nm. The maximum error range for measuring spectral parameters was 0.3%. It is worth noting the characteristic increase in reflection for wavelengths from 1 μm, which can be traced for all samples. Moreover, after laser treatment, this growth was not as noticeable as in the case of chemical texturing. Such an increase in the reflection coefficient can be associated with a sharp decrease in the intrinsic absorption of the silicon in the near-IR region of the spectrum [37]. In addition, the surface treated only chemically had a significantly higher reflection (on average, 25% more) in the short-wavelength region of the studied spectrum (from 250 nm to 400 nm) in contrast to the laser-textured surface. Such a striking difference between the reflection coefficients of surfaces processed by different methods can be explained by differences in the morphology of the side faces of the resulting microstructure. When using a laser in processing, the side face of the cones turns out to be uneven, and with chemical treatment, these irregularities are absent [23,25].

The absorptance in the studied spectral range was 95.6%, 84.9% and 92.1% for samples #1, #2 and #3, respectively. In this case, the greatest absorptance for the samples fell on the same wavelengths at which the minimum reflection was achieved, and amounted to 96.5%, 91.1% and 93.1% for three samples, respectively. Thus, the presence of laser processing in the formation of surface structures on the samples provides greater optical absorption (by about 7–10%), in contrast to the use of only chemical methods.

## 4. Conclusions

In this study we managed to produce light-trapping structures on silicon surface by various processing methods—laser, chemical and hybrid chemical/laser texturing. The chemical etching-based microtexturing of the samples consisted in successive immersion of silicon wafers into baths with reagents. Laser ablative texturing occurred during scanning of the surface of the samples by the focused femtosecond laser beam. As a result, the laser texturing of the preliminarily chemically etched Si wafers took five-fold less time than using the only laser texturing of the bare flat Si wafer.

Subsequently, the surface topography of the obtained structures was characterized, indicating its crystalline order, and the residual total reflectance of the textured silicon surfaces was measured. The resulting structure after the only laser texturing is a chaotic conical relief, and after chemical it is a chaotic pyramidal one. After both these processing procedures, the material remained in the polycrystalline state, as before processing. In this case, the hybrid-treated samples exhibited the average total reflectance in the spectral range of 250–1100 nm lower by 7–10%, than after the only chemical treatment.

## Figures and Tables

**Figure 1 materials-16-02350-f001:**
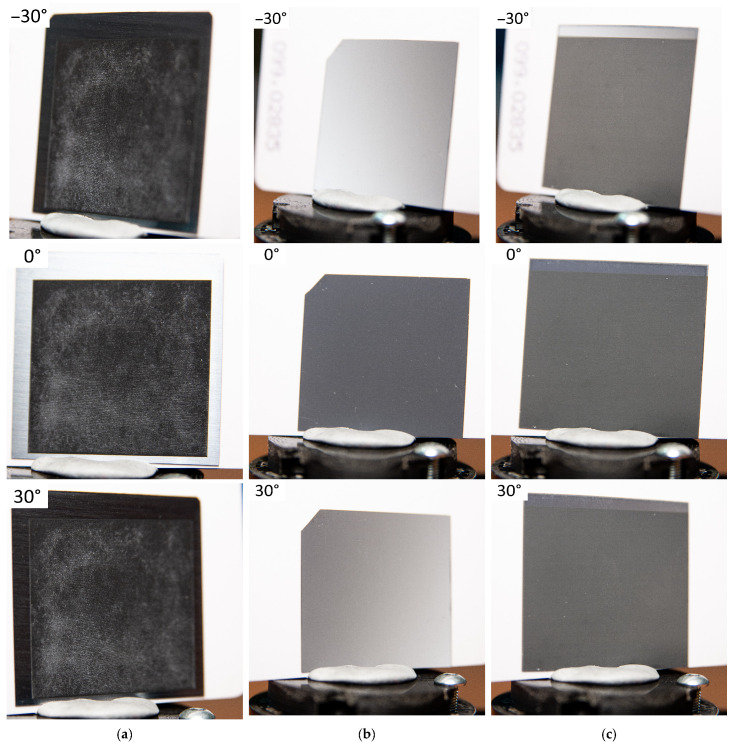
Optical images of processed areas of the silicon surface: (**a**) Sample #1—laser texturing of a silicon cut; (**b**) Sample #2—chemical etching; (**c**) Sample #3—hybrid chemical/laser fabrication.

**Figure 2 materials-16-02350-f002:**

SEM images of structures: (**a**) Sample #1—laser texturing of a silicon cut; (**b**) Sample #2—chemical etching; (**c**) Sample #3—hybrid chemical/laser fabrication.

**Figure 3 materials-16-02350-f003:**
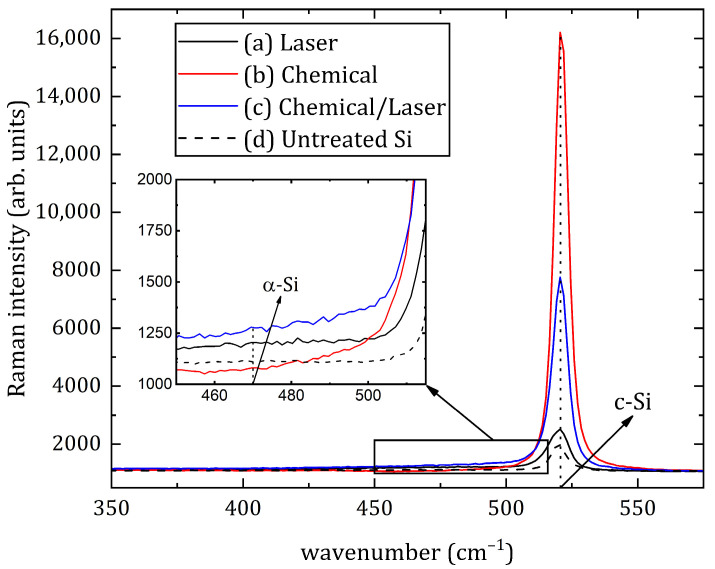
Raman spectra of silicon surface: (**a**) Sample #1—laser texturing of a silicon cut; (**b**) Sample #2—chemical etching; (**c**) Sample #3—hybrid chemical/laser fabrication; (**d**) Untreated Si.

**Figure 4 materials-16-02350-f004:**
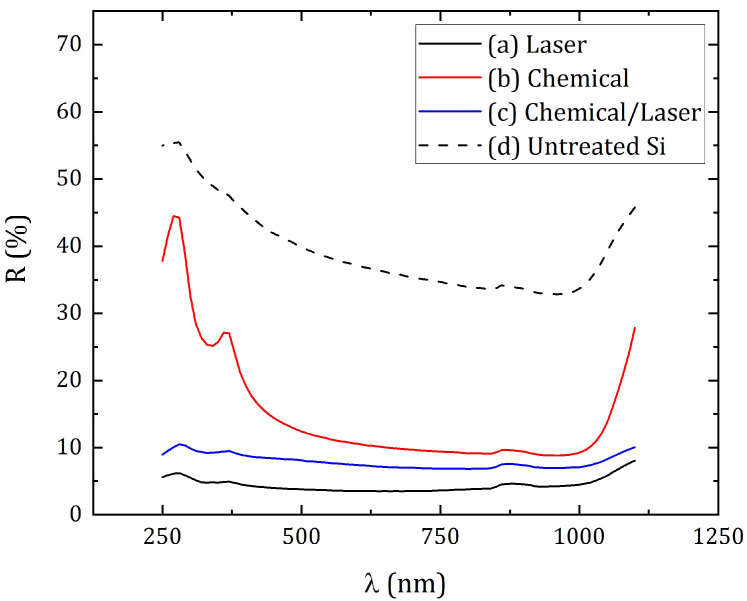
Total reflectance of silicon surface: (**a**) Sample #1—laser texturing of a silicon cut; (**b**) Sample #2—chemical etching; (**c**) Sample #3—hybrid chemical/laser fabrication; (**d**) Untreated Si.

**Figure 5 materials-16-02350-f005:**
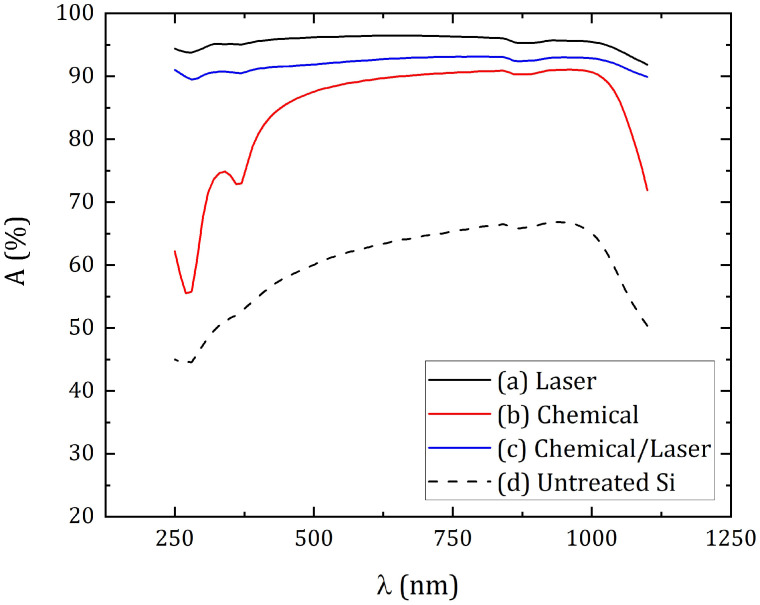
Absorptance of silicon surface: (**a**) Sample #1—laser texturing of a silicon cut; (**b**) Sample #2—chemical etching; (**c**) Sample #3—hybrid chemical/laser fabrication; (**d**) Untreated Si.

**Table 1 materials-16-02350-t001:** The numbers of samples and corresponding processing methods.

Sample #1	Sample #2	Sample #3
Laser microtexturing	Chemical etching	Hybrid chemical-laser processing

**Table 2 materials-16-02350-t002:** EDX atomic composition analysis of three samples.

Element	Sample #1	Sample #2	Sample #3
Silicon	87.5%	98.1%	85.7%
Oxygen	12.5%	1.9%	14.3%

## Data Availability

The supporting data could be provided upon a reasonable request.

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
