# Peer review of "Efficient Broadband Light-Trapping Structures on Thin-Film Silicon Fabricated by Laser, Chemical and Hybrid Chemical/Laser Treatments"

_materials, 2023, doi:10.3390/ma16062350_

Round 1

Reviewer 1 Report

The utilization efficiency of energy has always been an important aspect in the fields of photovoltaic industry. The authors propose a method of combining chemistry with laser treatments to make silicon materials suitable for a wider range of optical wavelengths, which is very practical. The following modifications or adjustments are recommended.

1. The abstract is too simple, and the experimental results and comparative discussion should be added in this part.

2. In the part of Materials and Methods, it is suggested to describe the numbers of samples and corresponding processing methods in a Table, which may be more intuitive. The experimental process is also suggested to be described in schematic diagram with text descriptions.

3. In Fig. 2, please add the scale bar in the view of SEM, and delete other information below the figures. Please explain the reasons for different shapes or structures of materials, not just the description of phenomena.

4. In Fig. 3, please mark the crystalline and amorphous phases at the corresponding positions, respectively. In addition, are there oxides formed during the processing of the material?

5. How many times have the light absorbance of different wavelengths been measured on the sample surface under different treatment methods? What is the error range? It should be reflected in the specific experimental data.

Reviewer 2 Report

Board comments,

This paper is a study to increase the absorptivity of the silicon carbide wafer using surface engineering. This was carried out with several methods as, chemical etching, laser surface texturing and combination of both. The surface features of the samples were correctly assessed and the reflectivity was excellently evaluated. The researchers achieve to increase the material absorptivity through these methods.

This paper can therefore be published in the journal after considering the next comments;

The abstract should include some sentences about the material (SiC wafer), the application of the material and the problem to be solve. This can improve the dissemination of you paper.

Some laser parameters (e.g., beam quality, focal length, Rayleigh length, TEM and raw laser beam diameter) should be added in the 2. Materials and Methods part to improve the understanding of your experiments. 

The supplier of the chemical products should be incorporate in 2. Materials and Methods part to provide aa good supplier of chemical products.

The detail of the scanning  electron microscopy (e.g., beam spot, current tension, type of electrons and potential acceleration) should be included in the 2. Materials and Methods part to improve the understanding of your experiments.   

Time process should be replace by pulse number in line 124 "cusing spot, the processing time was reduced by a factor of 5 due to the high scanning speed," because this laser is pulsed. 

Laser Induce Periodic Structure Surface of Figure 2.a. should be commented because this is a typical nanostructures generated by ultra-shot laser. In addition, a picture at high magnification of these nanostructures should be carried out. 

The mechanism of these structures generation should be indicated owing to this increases the dissemination of the article.

Specific Comments

My specific comments are the following. 

To add reference in line 38 "of laser pulses, namely, their short duration and high peak power, is being actively explored [REF]."

To include reference in line 184 "intrinsic absorption of the material in the near-IR region of the spectrum [REF]. In addition, the sur-"

To incorporate reference in 189 line "faces of the resulting microstructure. When using a laser in processing [REF], the side face of the"

Round 2

Reviewer 1 Report

The authors have made appropriate revisions. It is recommended to accept and publish this paper.